# The Usefulness of Cell-Based and Liquid-Based Urine Tests in Clarifying the Diagnosis and Monitoring the Course of Urothelial Carcinoma. Identification of Novel, Potentially Actionable, *RB1* and *ERBB2* Somatic Mutations

**DOI:** 10.3390/jpm11050362

**Published:** 2021-04-30

**Authors:** Tadeusz Kałużewski, Grzegorz K. Przybylski, Michał Bednarek, Sławomir Glazar, Magdalena Grabiec, Adam Jędrzejczyk, Łukasz Kępczyński, Izabela Kubiak, Dorota Kucharska, Agnieszka Morel, Magdalena Owczarek, Marek Rożniecki, Jordan Sałamunia, Dominika Szewczyk, Jarosław Szwalski, Bogdan Kałużewski

**Affiliations:** 1Laboratory of Medical Genetics of the “Genos” Partnership—R&D Division, 91-033 Lodz, Poland; michalbednarek@genos.com.pl (M.B.); magrabiec@gmail.com (M.G.); lukasz.kepczynski@iczmp.edu.pl (Ł.K.); iza.kubiak@genos.com.pl (I.K.); genos_stronsko@op.pl (D.K.); agnieszka.morel@genos.com.pl (A.M.); magdalenaowczarek@genos.com.pl (M.O.); jordan.salamunia@genos.com.pl (J.S.); dominika.szewczyk@genos.com.pl (D.S.); b.kaluzewski42@gmail.com (B.K.); 2Department of Genetics, Polish Mother’s Memorial Hospital Research Institute, 93-338 Lodz, Poland; 3Institute of Human Genetics, Polish Academy of Sciences, 60-479 Poznan, Poland; grzegorz_przybylski@yahoo.com; 4MyGenome Laboratory, 60-461 Poznan, Poland; 5Department of Pathomorphology, District Hospital, 98-220 Zdunska Wola, Poland; s.glazar@szpital-zdwola.info; 6Department of Urology, District Hospital, 97-500 Radomsko, Poland; adam46xy@gmail.com; 7Lekarze Urolodzy Marek Rozniecki i Partnerzy, Non-Public Department of Urology, 98-100 Lask, Poland; rozmaryn68@gmail.com; 8CYTOPATH S.A., Histopathological Laboratory, 90-552 Lodz, Poland; szw2000@poczta.onet.pl

**Keywords:** bladder cancer, urinary cytology, FISH, Bladder EpiCheck, non-invasive urine tests, WGS, urinary carcinoma

## Abstract

Bladder cancer is one of the most common cancers in global statistics. One of the issues associated with this disease is the high incidence of cases with delayed diagnosis and what factors correlate with worse treatment outcomes. A possible reason for this may be the rather limited availability of non-invasive diagnostic tools. This short communication presents a case of a 68 year old male patient after an ineffective therapy, carried on for several years with symptoms commonly associated with prostate overgrowth that masked a carcinoma in situ of the urinary bladder. Implementation of several diagnostic techniques, including urine sediment cytology, immunocytochemistry, the fluorescence in situ hybridisation technique, the Bladder EpiCheck test and whole-genome sequencing, enabled the establishment of a correct diagnosis, implementation of appropriate treatment and provision of patient-friendly monitoring. The described case emphasises the usefulness of cell-based and liquid-based urine tests in bladder cancer diagnostic procedures.

## 1. Introduction

### 1.1. Background

In global statistics, bladder cancer (BC) is the sixth and seventeenth most common cancer in men and women, respectively [1]. In 2018, 7494 new cases of BC were recorded in Poland (5612 in men—ranking the 5th, and 1882 in women—ranking the 13th in malignancy incidence in Poland), out of whom 3973 died (3073 men and 900 women) [2]. In the European Union countries, the total mortality rate from urinary bladder cancer stabilised at the beginning of the 1990s of the previous century and has gradually been falling since then, except in Bulgaria, Poland and Romania. During the years 2005–2008, Poland ranked the first with regards to the mortality rate from urinary bladder cancer [3]. One possible reason could have been the limited access to urinary cytology, caused by financial and local administrative issues. That situation restricted urologists to cystoscopy and precluded the usage of sensitive non-invasive tests. Here we attempt to emphasise the usefulness of voided urine examinations in bladder cancer diagnostics, proven in the case of a male patient with coexisting benign prostate hyperplasia and a carcinoma in situ of the urinary bladder.

### 1.2. Patient’s Medical History

In the presented case, a 61 year old male, working for 48 years in the metal processing industry and a tobacco smoker (10 pack-years), reported lower urinary tract symptoms (LUTS) to his general practitioner, including urinary hesitancy, poor stream and recurrent signs of bladder inflammation. After ineffective primary treatment with furazidine, the patient was referred to a urologist who started therapy with tamsulosin. Due to the lack of any improvement, finasteride was added to the treatment regimen, as well as an antibiotic therapy with several different agents was implemented. The above-mentioned treatment lasted from 2013 to 2018 with a rather poor clinical outcome—the patient began to suffer from a gradually developing nocturia. In 2018, the patient turned up at the Genos Outpatient Clinic where transrectal ultrasonography (TRUS) revealed an exophytic discreet lesion in the bladder triangle. Non-invasive urine tests were applied. The results of the performed diagnostics and the outcome of an introduced treatment are described below.

## 2. Materials and Methods

### 2.1. Cytourofish^(+)^

The Cytourofish^(+)^ is a comprehensive test consisting of urinary sediment cytology and fluorescence in situ hybridisation (FISH) procedure, using a tricolour molecular probe, specific for the 9p21 chromosomal region and for the centromeres of chromosomes 9 and 17. Depending on the initial test results (cytology plus FISH), the reactions of immunocytochemical expression are carried out with chosen antigens out of: p53, Ki67, CK20, or PCNA.

#### 2.1.1. Urinary Cytology

A specimen was obtained with the use of the “CYTOSET” cytology set (MPW MED. INSTRUMENTS, Warsaw, Poland). The routine Papanicolaou staining procedure was carried out without any modifications. The urinary cytology was evaluated in accordance with The Paris System for Reporting Urinary Cytology [4].

#### 2.1.2. FISH

A FISH examination was performed in accordance with an internal laboratory procedure (see Appendix A). The scope of normal values: a loss of the 9p21 region <5/100 cellular nuclei, the polysomy of chromosome 9 and chromosome 17 as the sum of <5/100 cellular nuclei. A “custom-made” probe was designed and set up by the PanPath Company (Budel; The Netherlands). 

#### 2.1.3. Immunocytochemical Staining

In the presented case, cytochemical reactions were done in accordance with an internal laboratory procedure (see Appendix B). The results of p53 staining were expressed as a percentage with a cut-off value, established at 50%. 

### 2.2. Bladder EpiCheck

The Bladder EpiCheck (Nucleix; Rehovot, Israel) test is based on an analysis of the panel of 15 DNA (deoxyribonucleic acid) biomarkers, associated with urinary bladder carcinoma. The test requires small amounts of DNA, isolated from easily available urine samples, while being simultaneously characterised by very high, clinically proven sensitivity, specificity and negative predictive value [5]. The Bladder EpiCheck test has been designed for the non-invasive monitoring of malignant relapses. The Bladder EpiCheck test enables the determination of the EpiScore, which is a measure of the overall biomarker panel methylation level, ranging from 0 to 100. The test cut-off is the EpiScore of 60, meaning that all the results, either equal to or above 60, are considered positive, while those below 60 are considered negative. The Bladder EpiCheck test was performed on a Rotor Gene Q instrument (Qiagen; Venlo, The Netherlands), following the kit operating manual without any modifications.

### 2.3. DNA Isolation and Whole Genome Sequencing (WGS)

DNA was extracted from urine sediment and whole blood samples. DNA isolation was performed using a commercially available Sherlock AX kit (A&A Biotechnology, Gdansk, Poland), according to the manufacturer’s instructions. The purity and concentration of extracted DNA samples were assessed using an Eppendorf BioSpectrometer^®^ basic (Eppendorf, Hamburg, Germany). The DNA concentration was 426.9 µg/mL for blood and 76.9 µg/mL for urine samples. Whole-genome sequencing of both sample types was performed at the Beijing Genome Institute (BGI, Hong Kong) by Next Generation Sequencing (NGS).

### 2.4. Bioinformatic Analysis

The analysis of WGS results was performed by means of bioinformatics tools, as well as by manual evaluation in the Integrative Genomics Viewer (IGV) browser v2.7.2 (Broad Institute, University of California) [6]. The identification of single nucleotide variations, short insertions and deletions, as well as discrimination of germline/somatic origin of mutations was performed by the VarScan v2.4.4 program [7] after data conversion to the mpileup format by the SAMtools v1.10 software [8]. The structural variant (SV) calling was performed with Delly [9] in case of translocations and bamCompare function of deepTools v2.0 [10] in case of deletions and amplifications. The graphic presentation of obtained data was visualised in the perl environment with the Circos v.0.69.9 software [11].

## 3. Results

### 3.1. Cytourofish^(+)^

In multifocal urothelial carcinoma in situ (CIS), neoplastic cells are characterised by the reduction of cohesion forces, which makes them easier to identify by urine sediment cytology, comparing to other cancers of the urothelial tract [12]. An analysis of cytological specimens enables to identification of high-grade urothelial carcinoma (HGUC), based on The Paris System for Reporting Urinary Cytology (see Figure 1).

The FISH procedure enables detection of cells with genomic alternations in the 9p21 region. A homo- (24% of examined cells) and heterozygous (60% of examined cells) losses in 9p21 locus were found. No numerical aberrations were detected by the probes specific for chromosome 17 and 9 centromeres. In the studied case, the immunocytochemical expression of the p53 gene was also examined; its presence was found out in 62% of urine sediments (with nuclear reaction in 40% and cytoplasmic reaction in 22%). A clearly positive reaction, i.e., above 50%, allows for differentiating neoplastic cells of the urothelium with reactive atypia (see Figure 2).

### 3.2. Bladder EpiCheck

In the presented case, the EpiScore value was 92, which represented a positive result of the Bladder EpiCheck test.

### 3.3. Whole Genome Sequencing (WGS)

#### 3.3.1. Structural Variations (SV)

Somatic SV’s can be defined as copy number alternations (CNAs), large deletions and amplifications, as well as chromosomal inversions and translocations. Bladder cancer usually contains around 300 SV’s per case [13]. Particular attention was paid to the most common alternations, observed in BC samples [13] (see Table 1). In the presented case, we managed to recognise 338 corresponding sequences within the genome and a variety of amplifications and deletions (see Figure 3 and Figure 4). A coverage analysis of chromosome 9, with the resolution levels of 1 kb and 10 kb, confirmed the 9p21 deletion, detected originally by Cytourofish^(+)^. Additionally, WGS revealed a deletion of the first 22 Mb of its short arm (9p22–24). The deleted region contained over 300 genes, including cyclin dependent kinase inhibitor 2A gene (*CDKN2A*) and SWI/SNF related, matrix associated, actin dependent regulator of chromatin, subfamily a, member 2 gene (*SMARCA2*). In addition, a 900 kb amplification was found at the position of 32.9 M–33.9 M that was rearranged to an amplified region of chr10, at the position of 35.5 Mb. That rearrangement, corresponding to unbalanced translocation t(9;10)(p13;p11), juxtaposed the serine protease 3 (*PRSS3*) and the ankyrin repeat domain 30A (*ANKRD30A*) genes. Another rearrangement, corresponding to unbalanced translocation t(4;17)(q31;q24), disrupted the transmembrane 131 like (*TMEM131L*) and the solute carrier family 39 member 11 (*SLC39A11*) genes, by fusing head to head the first 3 exons of *TMEM131L* and the first 5 exons of *SLC39A11*. The attention was also drawn by a lot of corresponding sequences between 1q23.3–24.1 and 11q13.3. 

#### 3.3.2. Single Nucleotide Variations, Deletions and Insertions

Variant calling enables the presentation of numerous single nucleotide variations (SNVs), short insertions (ranging from 1 bp to 26 bp) and deletions (ranging from 1 bp to 38 bp). All of the alternations present in the coding sequences of the patient genome are presented on a circular graph (see Figure 5). The chosen genes, frequently affected in bladder cancer, were additionally checked with the IGV browser (please refer to Appendix C for a complete list of manually analysed genes). 

The most clinically relevant finding was the somatic missense pathogenic variant of the tumour protein p53 gene (*TP53*) c.524G>T (p.Arg175Leu) that was associated with loss of heterozygosity (LOH). Further, the following unreported somatic heterozygous missense variants were found: erb-b2 receptor tyrosine kinase 2 gene (*ERBB2*) c.2180G>C (p.Gly727Ala), RB transcriptional corepressor 1 gene (*RB1*) c.2726C>T (p.Thr909Ile) and *RB1* c.1054G>C (p.Glu352Gln), and a somatic heterozygous synonymous variant of alpha-L-iduronidase gene (IDUA) c.2064G>A (p.Lys688=). Additionally, we recognised associated with LOH germline missense variants: lysine demethylase 6B gene (*KDM6B*) c.2917A>C (p.Lys973Gln) and syntaxin binding protein 1 gene (*STXBP1*) c.1127C>T (p.Thr376Ile), and associated with LOH germline synonymous variants of latent transforming growth factor beta binding protein gene 2 (*LTBP2*) and actin-dependent regulator of chromatin, subfamily a, member 4 gene (*SMARCA4*). We also identified germline heterozygous missense variants of: nuclear receptor corepressor 1 gene (*NCOR1*) c.59A>C (p.Tyr20Ser) and *IDUA* c.1554A>C (p.Leu518Phe) (see Table 2).

### 3.4. Further Proceeding

With the above-mentioned diagnostic data in hand, cystoscopy was carried out, undertaking a non-random bladder mucosa biopsy. Four specimens were collected, identifying urothelial carcinoma (HG CIS ICD-O code 8120/2) in each localisation and cystitis cystica, which could be responsible for the change seen in TRUS (see Figure 6 and Figure 7).

It was decided to apply local immunotherapy with intravesical injections of attenuated mycobacteria (bacillus Calmette-Guérin—BCG-Medac, medac GmbH, Wedel, Germany). At the moment, the patient is undergoing the maintenance immunotherapy, according to the scheme proposed by Lamm et al. [14]. Demonstrating no disease-related symptoms, the patient returned to work, while remaining under further medical follow-up with the periodical application of non-invasive diagnostic tests. Since the application of treatment, the patient has not presented either LUTS or any other of the previously identified symptoms.

### 3.5. Non-Invasive Monitoring

The European Association of Urology recommends base follow-up of TaT1 tumours and carcinoma in situ (CIS) via regular cystoscopy [15]. In the presented case, the patient agreed to simultaneous non-invasive monitoring with urine cytology (see Figure 8) and the Bladder EpiCheck test (see Table 3). The above-mentioned results were consistent with the invasive monitoring, based on blue light cystoscopies, with histopathological assessment (see Figure 9), as well as with the clinical picture.

## 4. Discussion

In the presented case, the rather misfortunate circumstance of a 5-year long diagnostic process was mainly caused by the lack of easily accessible urinary cytology and the concomitance of two medical conditions. The LUTS, that can barely occur as a bladder cancer manifestation, prompted the diagnostic process towards isolated benign prostate hyperplasia. It is worth noting that the patient did not present erythrocyturia, a typical bladder cancer symptom. The white light cystoscopy could not reveal the carcinoma in situ within the bladder mucosa, which follows from the tumour morphology. It is reflected by a low sensitivity of this technique in that clinical condition [16]. In the described case, the test that finally led to biopsy of the bladder was urinary cytology that revealed a high-grade urothelial carcinoma. The reason for a simultaneous extension of the diagnostics with Cytourofish(+) resulted from our long-term experience with the application of the UroVysion^®^ test [17]. The FISH technique, as well as immunocytochemistry, supported the initial diagnosis. The remaining tests were performed for cognitive, not diagnostic purposes.

A comparative analysis of blood and urine DNA samples, performed with whole-genome sequencing, enabled to presentation of the genomic status of the tumour with multiple changes, which could potentially lead to a more accurate classification of the disease and reveal possible alternative therapeutic options. It should be emphasised that the isolation of DNA from urine is technically easier and results in better DNA quality than the commonly used FFPE (Formalin-Fixed Paraffin-Embedded) samples (our experience, data not published). It should be taken under consideration in the process of implementation of genomics sequencing into clinical practice. During the analysis we were able to recognise several single nucleotide variants, including a pathogenic variant of *TP53* gene that may have contributed to the cancer formation. *ERBB2* mutations are known to be present in bladder cancer and could be potentially used for precision medicine purposes. They are not mutually exclusive of gene amplification [18]. The patient was successfully treated with injections of attenuated mycobacteria, as described above, however, the whole genome sequencing, that enabled recognition of the amplification of *ERBB2* gene, potentially entitles therapy with trastuzumab, lapatinib and many more target-oriented therapies, according to mycancergenome.org. Taking into consideration the missense *ERBB2* mutation present in the specific population of cells in urine sediment, the relevance of amplification should be confirmed, for example, by fluorescence in situ hybridisation with the use of an HER2/neu molecular probe. While the above-mentioned therapy is not popular in bladder cancer cases (three phase 2 clinical trials opened on the date of the manuscript edition), we could not claim that the patient would not receive any benefits from it. The crucial role of *RB1* gene in oncogenesis is commonly known [19]. However, the significance of variants, recognised in the presented case, still remains to be determined. 

Weinstein et al. [20] proposed a novel bladder cancer classification based on the molecular events in the tumour DNA. Three groups (A, B and C) are described in the following way: Group A—the amplifications of *E2F2/SOX4*, *EGFR*, *PPARG*, *PVRL4*, *YWHAZ*, *MYC* and the mutations of *MLL2* gene; Group B—the deletion of *CDKN2A* and the activating mutations of FGFR3; Group C—*TP53* and *RB1* mutations, the amplifications of *E2F3* and *CCNE1*. Despite the fact that some elements, which form each of the above-mentioned groups, were present in the evaluated material, we could make an attempt to qualify the patient to Group C, while both the point mutations and amplifications were present in the DNA from urine sediments. The analysis of genetic events also enabled the reconstruction of the biological pathways of the tumour development. Relying on our findings, we could recognise disruptions in the TP53/RB1 pathway, as well as disturbances in chromatin remodelling. Some elements of the RTK/RAS/PI(3)K pathway were also altered. The recognition of *TP53* mutation and thus belonging to group C resulted in very probable resistance to a cisplatin-based chemotherapy [21], which may be a crucial information for future personalisation of patient therapy. A complete characterisation of molecular events in the presented case can be found in Appendix D. 

The invasive procedure of cystoscopy with histopathological evaluation confirmed the initial diagnosis and enabled the introduction of an effective treatment. While the benefits and limitations of monitoring the high-grade urothelial tumours with cystoscopy and cytology are commonly recognised [15], it was possible in the presented case to use the Bladder EpiCheck test. The test was done before and in the course of the therapy. The obtained results were consistent with the invasive monitoring, based on cystoscopies, as well as with the clinical picture. The presented case provides further evidence for the relevancy of the monitoring schedule, based on switching between cystoscopy and the Bladder EpiCheck test, as proposed by the Radbound University Medical Center [22]. 

The new epoch of personalised medicine has been creating new demands for diagnostic processes. It is hard to tell if the performed actions were optimal for the patient in the presented case report. Nevertheless, the amount of facts gained from the non-invasive urine tests emphasises their usefulness and may contribute to future diagnostic protocols and therapeutic scenarios. Especially their susceptibility to machine learning algorithms opens a wide range of potentially effective clinical applications.

## Figures and Tables

**Figure 1 jpm-11-00362-f001:**
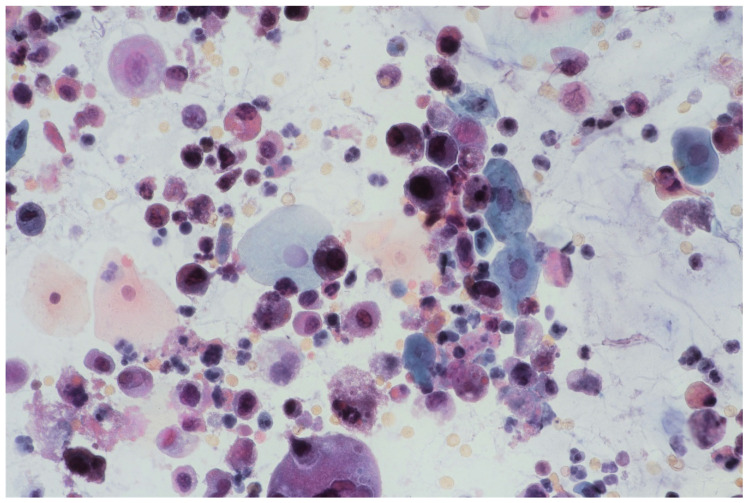
Urine sediment—a cytological specimen. Papanicolau staining. Urothelial carcinoma cells with a high N/C (nucleus/cytoplasm) ratio (~30% of cells with N/C ratio ≥0.7 and ~20% with N/C ratio ≥0.5 and <0.7), nuclear margin irregularity and hyperchromasia. Prominent single or multiple nucleoli and mitoses. An inflammatory background of the specimen is visible with numerous granulocytes and erythrocytes.

**Figure 2 jpm-11-00362-f002:**
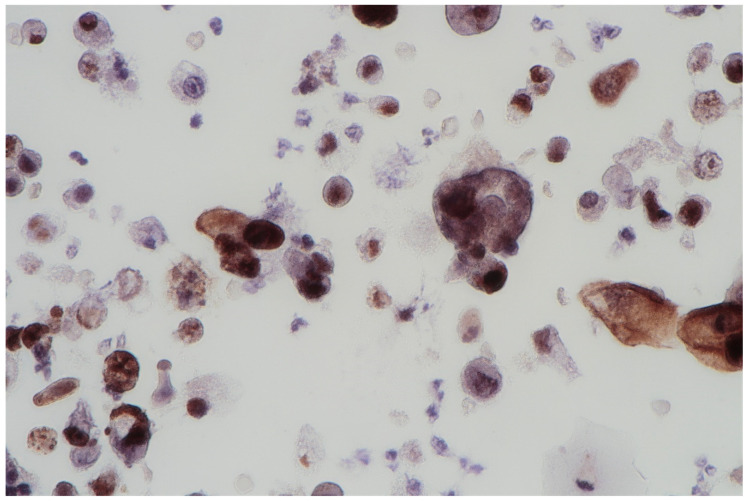
Urine sediment—a cytological specimen. The immunocytochemical reaction of p53. A dominating picture of positive p53 reactions, both nuclear and cytoplasmatic. Single large cells are filled with haemosiderin.

**Figure 3 jpm-11-00362-f003:**
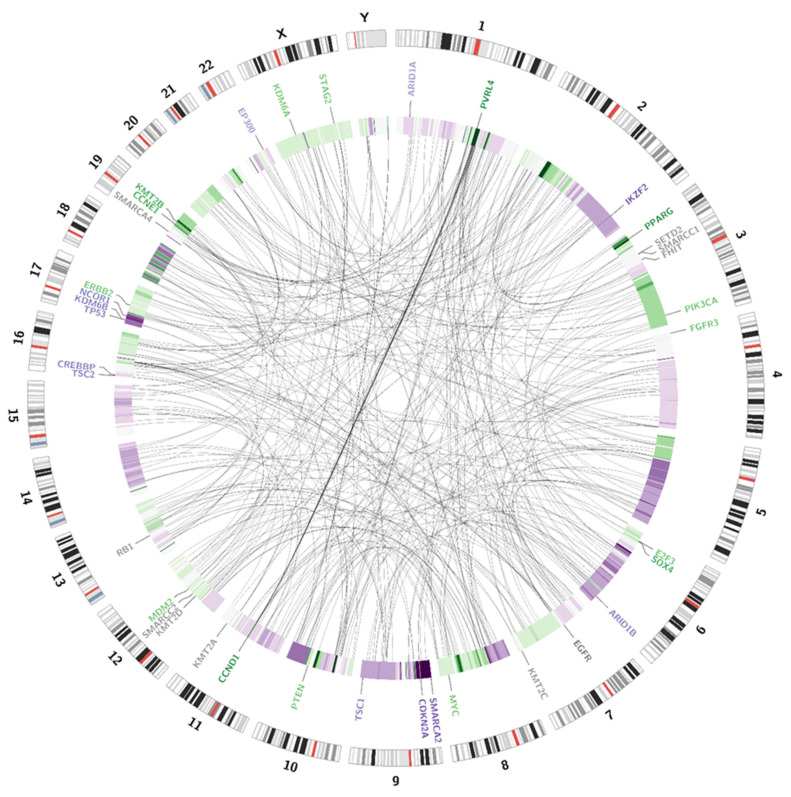
Visualisation of structural variations. The external circle represents an ideogram, while the internal circle illustrates the coverage difference between the patient’s blood samples and urine samples. The darker the green area, the larger are the amplification rates. The darker the purple colour, the larger is the deletion. Genes are coloured according to the copy number alternations of their loci: purple—deletions; green—amplifications; grey—no change in copy number. The link inside the graph visualises the pattern of corresponding sequences.

**Figure 4 jpm-11-00362-f004:**
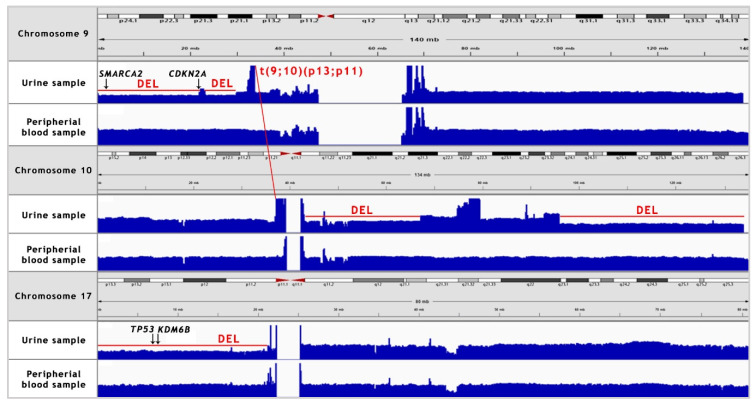
NGS coverage visualisation of the copy number variations (CNV’s) of chromosomes 9, 10 and 17 in the urine sample, as compared to the peripheral blood sample. A manual analysis enabled to confirm the deletion of 9p21 locus. Additionally, 9p22–24 deletion, two major deletions in long arm of chromosome 10, 17p deletion and translocation between chromosomes 9 and 10 were recognised. “DEL” stands for deletion.

**Figure 5 jpm-11-00362-f005:**
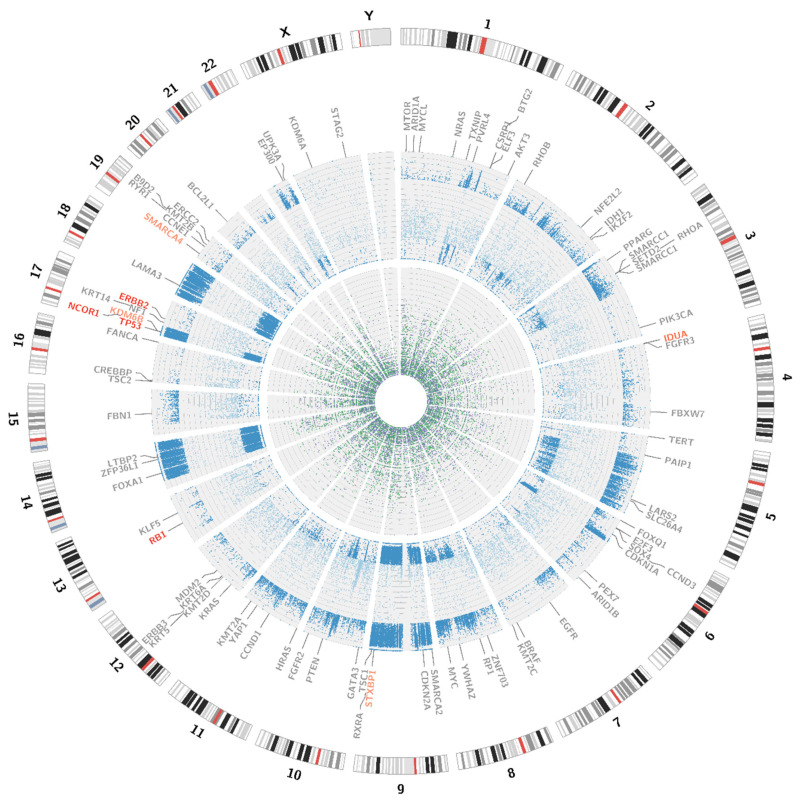
Visualisation of SNPs, insertions and deletions. The external circle represents an ideogram, while the internal circles represent the distribution of mutations. The dark blue dots stand for the loss of heterozygosity events; the light blue dots for novel mutations. The green dots represent insertions and the purple dots represent deletions. All the genes, analysed for the presence of somatic mutations are visible on the graph, the affected genes are marked red.

**Figure 6 jpm-11-00362-f006:**
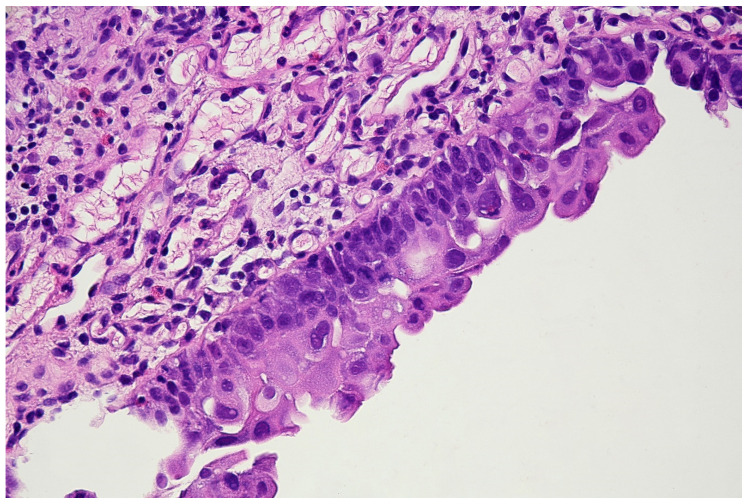
Histopathological specimen—Haematoxylin and Eosin (H&E) staining. The bladder is covered with the epithelium, the cells of which show features of severe atypia. Atypical cells are visible in all the layers of the epithelium, including the external surface. Locally, a reduced number of cell layers can be seen. Nuclei are present with marked anisokaryosis, hyperchromasia, and sometimes, with irregularity of the nuclear membrane. The unbroken basement membrane is visible along the entire length of the epithelium; there is no invasion of atypical cells. Occasionally, mitoses with atypical division figures can be noted.

**Figure 7 jpm-11-00362-f007:**
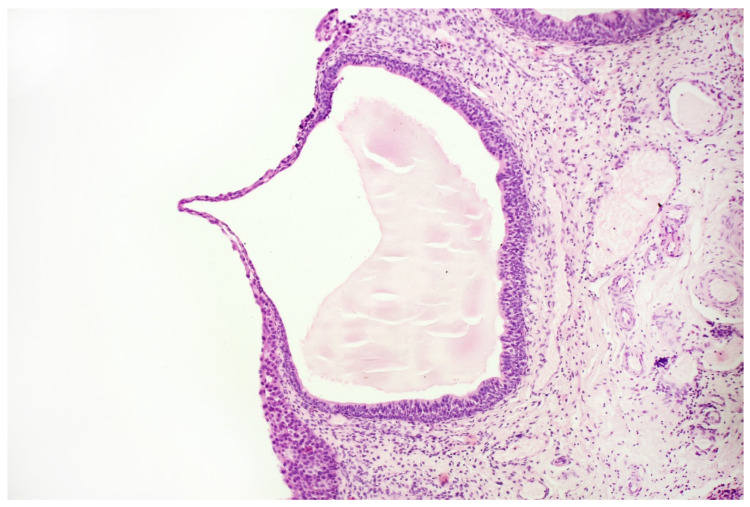
Histopathological specimen—Haematoxylin and Eosin (H&E) staining. Cystitis cystica. The cysts are filled with homogeneous eosinophilic fluid and lined with the urothelial epithelium, the cells of which do not show signs of atypia. The atypia is visible in the epithelium, lying on the surface of the bladder in the immediate vicinity of the described cystic structures, in the submucosa of the bladder. There is no crawling of the atypical epithelium into these cysts.

**Figure 8 jpm-11-00362-f008:**
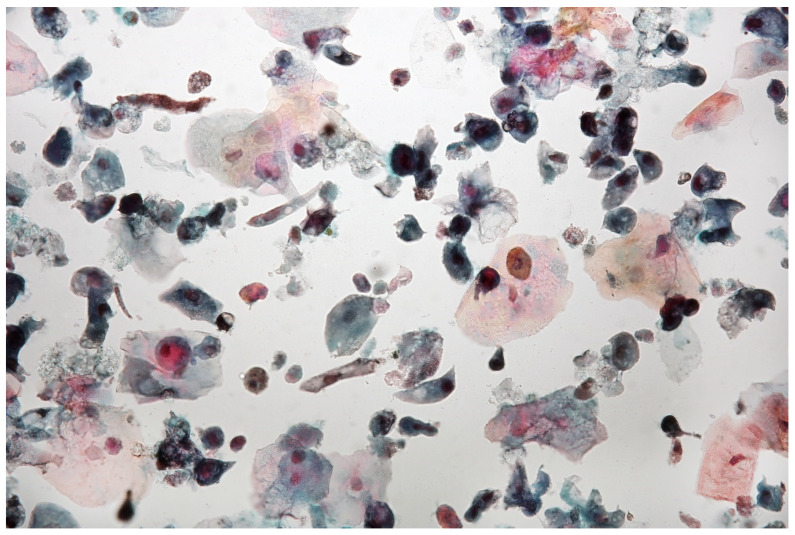
Urine sediment—a cytological specimen. Papanicolau staining. Atypical cells with high N/C ratios (~5% of cells with N/C ratio ≥ 0.7 and ~12% with N/C ratio ≥0.5 and <0.7) are visible. A significant amount of normal urothelial cells are visible. There are a few multinucleated cells, granulocytes and uric acid crystals. Significant vacuolisation of the cellular cytoplasm may indicate a strong regenerative reaction. The AUC (atypical urothelial cells) result was stated, based on The Paris System for Reporting Urinary Cytology.

**Figure 9 jpm-11-00362-f009:**
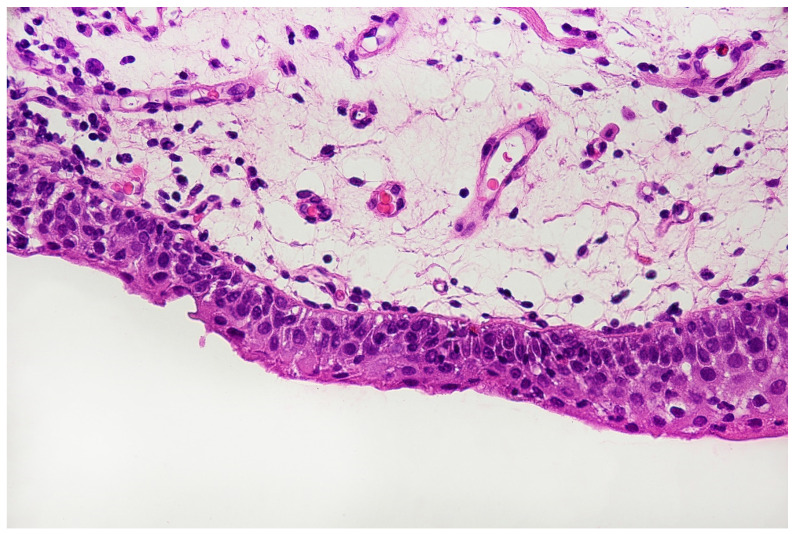
Histopathological specimen—Haematoxylin and Eosin (H&E) staining. The bladder mucosa with signs of hyperaemia and numerous thin-walled blood vessels are visible. Moderately severe inflammatory infiltrations are present with monocular leukocytes. The section is covered with a metaplastic epithelium with the appearance of a flat epithelium. The number of layers of the epithelium is reduced, the cells show no signs of atypia.

**Table 1 jpm-11-00362-t001:** Most frequent amplifications and deletions, observed in the bladder cancer, and their presence in the described case.

Amplifications		
Chromosomal Locus	Gene	Presence in the Evaluated Case
6p22.3	*E2F3*	Yes
6p22.3	*SOX4*	Yes
11q13.3	*CCND1*	Yes
7p11.2	*EGFR*	-
17q12	*ERBB2*	Yes
19q12	*CCNE1*	Yes
3p25.2	*PPARG*	Yes
12q15	*MDM2*	Yes
8q24.21	*MYC*	Yes
4p16.3	*FGFR3*	Yes
1q23.3	*PVRL4*	Yes
3q26.32	*PIK3CA*	Yes
**Deletions**		
9p21.3	*CDKN2A*	Yes
2q34	*IKZF2*	Yes
13q14.2	*RB1*	-
3p14.2	*FHIT*	-
16p13.3	*CREBBP*	Yes
17p12	*NCOR1*	Yes
12q13.12	*KMT2D*	-
11q23.3	*KMT2A*	-
7q36.1	*KMT2C*	-
19q13.12	*KMT2B*	-
22q13.2	*EP300*	Yes
1p36.11	*ARID1A*	Yes
6q25.3	*ARID1B*	Yes
9p24.3	*SMARCA2*	Yes
19p13.2	*SMARCA4*	-
3p21.31	*SMARCC1*	-
12q13.2	*SMARCC2*	-
9q34.13	*TSC1*	Yes
16p13.3	*TSC2*	Yes
10q23.31	*PTEN*	-
3p21.31	*SETD2*	-
17p13.1	*KDM6B*	Yes
Xp11.3	*KDM6A*	-
Xq25	*STAG2*	-
17p13.1	*TP53*	Yes

**Table 2 jpm-11-00362-t002:** Selected mutations with a potential impact on bladder cancer development. Clinical significance based on varsome.com.

Gene	Variant	Percentage of Reads Blood Sample	Percentage of Reads Urine Sample	Clinical Significance
*TP53*	c.524G > T (p.Arg175Leu)	0%	78%	Pathogenic (LOH)
*ERBB2*	c.2180G > C (p.Gly727Ala)	0%	28%	Likely pathogenic
*RB1*	c.2726C > T (p.Thr909Ile)	0%	33%	Uncertain significance
	c.1054G > C (p.Glu352Gln)	0%	36%	Uncertain significance
*KDM6B*	c.2917A > C (p.Lys973Gln)	59%	89%	Benign (LOH)
*STXBP1*	c.1127C > T (p.Thr376Ile)	50%	93%	Uncertain significance (LOH)
*IDUA*	c.1554A > C (p.Leu518Phe)	14%	29%	Uncertain significance
*SMARCA4*	c.2064G > A (p.Lys688=)	0%	38%	Benign
	c.4680C > T (p.Asp1560=)	35%	74%	Likely benign (LOH)
*LTBP2*	c.2502T > C (p.Thr834=)	48%	89%	Benign (LOH)
*NCOR1*	c.59A > C (p.Tyr20Ser)	14%	29%	Uncertain significance

**Table 3 jpm-11-00362-t003:** Results of monitoring with the Bladder EpiCheck test. The follow-up tests began after the initial six doses of BCG treatment.

Primary Result	Follow-Up Result after 3 Months	Follow-Up Result after 6 Months	Follow-Up Result after 12 Months
Positive, EpiScore: 92	Negative, EpiScore: 21	Negative, EpiScore: 17	Negative, EpiScore: 7

## Data Availability

The data presented in this study are available on request from the corresponding author.

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
