# Peer review of "The Usefulness of Cell-Based and Liquid-Based Urine Tests in Clarifying the Diagnosis and Monitoring the Course of Urothelial Carcinoma. Identification of Novel, Potentially Actionable, RB1 and ERBB2 Somatic Mutations"

_jpm, 2021, doi:10.3390/jpm11050362_

Round 1

Reviewer 1 Report

The authors described a case in which they could diagnose bladder cancer effectively using recent non-invasive techniques. The results of the tests were consistent with conventional findings of bladder cancer and the usefulness of these tests in the diagnostic and follow-up period was suggested. Nevertheless, performed tests seem to be overscreening for the patient at that time.

Broad comments

While the results of recent techniques were interesting, this case doesn’t seem to be suitable as a case report due to the problems with clinical assessment. More valuable information for readers should be needed.

The author’s assessment algorithm of refractory LUTS or suspected bladder cancer was not in a general way. If bladder cancer had been suspected, a random biopsy based on the positive finding of urine cytology might be sufficient according to several guidelines. The author needs to describe the validity of the other tests more clearly.

Specific comments

  1. Some abbreviations were used without a full name.
  2. Common name of the drug (finasteride?) should be used.
  3. ???? (Line. 123)

Author Response

We thank you very much indeed for your time devoted to our manuscript and for your valuable comments. We would now like to refer to them, while giving some explanations.

Broad comments:

We agree with the opinion that, in the presented case, we performed a much wider diagnostics than would have been sufficient to obtain a reliable diagnosis. It is emphasized in Discussion that urine cytology was really enough to justify random biopsy of the bladder (lines 255-257). We need here to highlight the fact that is involved in the diagnostics of bladder cancer in our country. The National Health Fund does not refund either urine sediment cytology, tests based on molecular cytogenetics (e.g. UroVysion), or other non-invasive tests, supporting the diagnosis of bladder cancer. The lack of availability of those methods was the reason why our diagnostics lasted 5 years. In our country, the diagnostic apparatus applied by us is fairly innovative and we thus wanted to make a wider group of urologists aware of its advantages. It is a known fact that complementing cytology with FISH and/or immunochemistry can increase the sensitivity of routine cytology. We would also like to point out that the next-generation sequencing was performed for cognitive, non-diagnostic purposes (we have added this sentence in discussion to make it clear for the readers, lines 259-260). While performing the Bladder EpiCheck test, we tried to find out its clinical usefulness in disease monitoring by reference to the proposed schemes (what was explained in Discussion, lines  296-304).

Specific comments:

  1. We have provided full names of most important genes. All of the abbreviations are now used with a full name given in the manuscript.
  2. We have changed the Latin name to English name (finasteridum > finasteride).
  3. The typing errors have been deleted.

Enclosed, please find a revised version of the manuscript, together with answers and comments for the Reviewers. The final version of manuscript has been proofread by a certified English translator.

Reviewer 2 Report

The paper entitled " The usefulness of cell-based and liquid-based urine tests in clarifying the diagnosis and monitoring the course of urothelial carcinoma. Identification of novel, potentially actionable, RB1 and ERBB2 somatic mutations.” is interesting. These types of studies should be encouraged for future. The paper is well written. However, some points could be clarified:

-Predictive model could be performed?

-how can be translated to clinical setting? Implementation?

-A resume integrative picture could be done to provide an illustration of data and transpose it.

Some typos could be reviewed.

Author Response

We thank you very much for your time devoted to our manuscript and your valuable comments. We would like now to refer to them in detail, providing our explanations.

  1. The diagnostic tools, as well as the therapeutic scheme, gave a surprisingly good outcome in the presented case. We tried to emphasise the utility of non-invasive tests and encourage their use in future diagnostic protocols and therapeutic scenarios. The information in the last paragraph of Discussion is, in our opinion, an attempt to encourage using the results in future, designed into a predictive model.
  2. We have added the section of clinical implementation according to molecular classification in the paper (lines 292-294). We believe that other findings have their clinical importance, well described in Discussion: another proof of the usefulness of urine cytology and FISH studies (lines 255-260), a potential treatment option in patients with ERBB2 alternation (line 271-274) and disease monitoring with the Bladder EpiCheck test (lines 296-304).
  3. We applied many different ways of data presentation and, the integrative picture of all the findings was a primary atempt. However, so many data in one picture was difficult to read in standard paper resolution. That is why we decided to split it to more pictures. We believe that the current method of data presentation should most suitable to the readers.
  4. The manuscript has been proofread by a certified English translator so it should not contain any typos.

Enclosed, please find a revised version of the manuscript with our answers and comments for all of the Reviewers. The final version of manuscript has been proofread by a certified English translator.

Reviewer 3 Report

The manuscript “The usefulness of cell-based and liquid-based urine tests in clarifying the diagnosis and monitoring the course of urothelial carcinoma. Identification of novel, potentially actionable, RB1 and ERBB2 somatic mutations” reports on the diagnosis tests for a patient. This report missed too many data (e.g., FISH) described in materials and methods. The authors performed NGS to identify the genomic alterations between peripheral blood and urine sediment but If the authors consider detecting mutations  of known genes (e.g., RB1 and ERBB2 in the title) from urine sediment is useful for diagnosis, classic target specific PCR and sequencing have advantage to NGS test.

Minor points

  1. Materials and methods have method for FISH analysis but no data was presented.
  2. Figs. 1, 2, and 8 showed cytology test from urine sediments with malignant cells. There were also other somatic cells such as white blood cells. It will be quite important to remove gnomic contaminants for the detection and quantification of tumor-specific genomic alteration observed in fig. 4. How the author excluded germline DNA contaminant to achieve mapping data of malignant cell?
  3. NGS mapping result (fig. 4) displays a characteristic deletion in chromosome 9p. Was it amplification of tumor genome or remained (undeleted) allele around CDKN2A?. Homozygous deletion and high copies around centromere of ch9q are also observed in germline genome from blood. Please account for these alterations. Chromosome 10q also displays urine-spechific amplification between deleted sequences. What genes there?
  4. Table 2 needs information of LOH. Although Fig. 4 showed one allele p53 deletion  in urine sediment cells, please show p53 status in blood DNA.

Author Response

We thank you very much for your time devoted to our manuscript and your valuable comments. We would now like to refer to them and give some more detailed explanations.

  1. The genes, included in the title were placed there after the study was completed. We did not know that we would discover novel mutations in RB1 and ERBB2 genes before analysis of NGS results. We agree that target specific PCR or Sanger sequencing may be beneficial, while looking for concrete mutations but it was not a case in our study. In the scope of looking for new mutations and genetic events there is no more cost-effective method that NGS in the current state of genetic laboratory diagnostics.
  2. The results of the FISH analysis were presented in the manuscript (lines: 124-127).
  3. In order to exclude the germline alternation, we performed a comparative analysis of urine and blood samples (described in Material and methods, line: 94 and in Discussion, line: 261). The discrimination of germline/somatic mutations was done with the use of the VarScan software. To make it more clear for the readers, we have added an information in “2.4. Bioinformatic analysis” section (line 105).
  4. The CDKN2A was entirely deleted but there is clearly visible higher number of reads right after the CDKN2A gene. There was a small normal biallelic region just downstream from CDKN2A. There is no proof that it was an amplification of tumor DNA. The CNVs around the centromere of ch9q are artifacts, generated by repetitive sequences that are abundant in the pericentromeric regions of the chromosomes. Since the NGS reads have the same sequence there are problems to properly align them to these regions of the genome. The lack of signals from the centromeres does not mean a deletion but informs that no reads could be mapped to these regions. Similar artificial CNVs in the pericentromeric regions were also observed in other individuals; please, see the attached Figure 1 (A: the urine sediment sample, B and C: unrelated individuals).

Indeed, chromosome 10 shows urine-specific amplification between the deleted sequences.

The first amplified region in chr10p of 1.8 MB (37.0M-38.8), mentioned in the manuscript because of its rearrangement to ch9, contains the following genes: ANKRD30A (the closest to the fusion region; mentioned in the manuscript), MTRNR2L7, ZNF248, ZNF25, ZNF33A, ZNF37A, HSD17B72P and ACTR3BP5.

  1. The second and the third amplified regions in chr10q of 12.5 MB (67.0M-79.5M) and 3.1 MB (92.8M-95.9M) contained many genes. Enclosed, please find Figures 2-4, showing zoomed in amplified regions with the names of the affected genes. We were able to recognize hundreds of urine specific amplifications like the ones in chromosome 10. It is not possible to analyse all in one manuscript. In our study, we focused on the regions that resulted in molecular classification, explained the pathway of tumour development or confirmed the results of other molecular tests. This is the reason why we did not present the genes, amplified in the region of chromosome 10 that you mentioned. However we appreciate the comment and we will consider it in our next study dedicated to detailed bladder cancer development, that we are currently realizing on much bigger number of cases.
  2. Table 2 has been changed in line with the Reviewers’ suggestion, we have now presented the mutation status in both urine and blood samples.

Enclosed, please find a revised version of the manuscript with our answers and comments for all of the Reviewers. The final version of manuscript has been proofread by a certified English translator.

Figure 1 Chr9 pericentromeric region

Figure 2 Chr10p amplification

Figure 3 chr10q amplification (67.M-79.5M)

Figure 4 chr10q amplification (92.8M-95.9M)

Round 2

Reviewer 1 Report

The authors have revised the manuscript more clearly and addressed concerns raised by reviewers. 

Reviewer 2 Report

 Authors have addressed my comments.

Reviewer 3 Report

The original manuscript jpm-1160549 entitled "The usefulness of cell-based and liquid-based urine tests in clarifying the diagnosis and monitoring the course of urothelial carcinoma. Identification of novel, potentially actionable, RB1 and ERBB2 somatic mutations." has been adequately revised in this version.   The author  appropriately answered for the review report and the points were clearly reflected in this version. 

This manuscript is a resubmission of an earlier submission. The following is a list of the peer review reports and author responses from that submission.

Round 1

Reviewer 1 Report

This is a case report of a carcinoma in situ (CIS) of the urinary bladder in which the authors say several diagnostic techniques, including urine sediment cytology, immunocytochemistry, FISH technique, Bladder EpiCheck test and whole genome sequencing, were useful to diagnose. For me, this was not interesting.

Major points

Delay of diagnosis

This patient had not got the correct diagnosis of bladder CIS for several years. If he had undergone conventional urine cytology in this period, he would have been diagnosed with it much earlier. Several guidelines including EAU guideline recommend urine cytology for the diagnosis of high-grade bladder cancer including CIS. I think urine cytology was enough for him.

Diagnostic techniques

No new diagnostic techniques were used in this study. And again, I do not think all of them were necessary for him. This is just a case report. If the authors want to show the usefulness of these tests, they should conduct clinical study.

Minor point

TRUS lesion

TRUS revealed an exophytic 5 × 4 mm 57 lesion in the bladder. I am wondering what it was because CIS could not generally been detected by ultrasonography.

Reviewer 2 Report

None

Reviewer 3 Report

The authors described a case in which they could diagnose bladder cancer effectively using recent non-invasive techniques. The results of the tests were consistent with conventional findings of bladder cancer and the usefulness of their tests in the diagnostic and follow-up period was suggested.

Broad comments

While the results of novel techniques were interesting, there are several issues to be revised.

  1. The result of urinalysis and the presence of hematuria should be provided.
  2. The author’s assessment algorithm of refractory LUTS was not in a general way. If bladder cancer had been suspected, a random biopsy based on the positive finding of urine cytology might be sufficient. The author needs to describe the validity of the other tests more clearly.
  3. It seems to be overscreening. Details for cost are not given in the discussion section.
  4. A description of misfortune and misdiagnosis may be nonsense because the onset date of bladder cancer is unknown.

Specific comments

  1. Several abbreviations were used without a full name. (e.g. GP, LBC, NGS, and CNV, etc.)
  2. Common name of the drug should be used. (Line 53)